# MORE SYSTEMATIC THAN CLAIMED: INSIGHTS ON THE SCAN TASKS

**Markus Kliegl & Wei Xu**
Institute of Deep Learning
Baidu Research
Sunnyvale, CA 94089, USA
{klieglmarkus,wei.xu}@baidu.com

## ABSTRACT

We show that some standard attention-based architectures widely used in Neural Machine Translation as well as a pointer-based variant achieve results on some of the compositional SCAN tasks that are far superior to those reported in Lake & Baroni (2018). We next show that there is high variance in the test accuracy across both random initialization and training duration. We show that ensembling can be used to take advantage of this variance and improve results but that, for many tasks, a large gap remains between ensemble performance and the performance of an oracularly selected single best model. Based on these insights, we suggest some possible directions for future research, emphasizing selection and regularization over the need for more compositional architectures.

## 1 INTRODUCTION

In Lake & Baroni (2018), several synthetic machine translation tasks were designed to test compositionality of sequence-to-sequence models. For example, a model that has been trained to translate "jump" and "walk thrice and turn around left" would be expected to also be able to translate "jump thrice and turn around left." The authors suggest that modern sequence-to-sequence architectures such as (attentional) encoder-decoder architectures do not behave systematically. We show in this paper that several widely used architectures are more systematic than claimed and that it may be more fruitful to focus on other issues highlighted by the SCAN tasks.

## 2 COMPARISON OF ATTENTION MECHANISMS

We compare two attention mechanisms, Bahdanau and Luong, that have been studied in Neural Machine Translation (Britz et al., 2017). For both of these attention mechanisms, we further compare the standard implementation with a pointer-based alternative, in which the attention mechanism is used to extract vectors from the input embedding rather than the encoder output sequence. The motivation for this is that the pointers should induce a positional inductive bias on the decoding that we would expect to generalize better on the SCAN tasks. We describe the attention mechanisms and the pointer architecture in more detail in Appendices C and D. All models were trained for 10,000 iterations at batch size 128. We test on the various SCAN tasks described in Appendix A. The full implementation details and hyperparameter settings are given in Appendix B.

The results are shown in Table 1. Especially noteworthy is the increase from 1.2 % to 14 % accuracy by the Ptr-Luong architecture on the `addprim_jump` task, though this is still far from where we could consider the task solved. The `simplified_length` task is a modification of the original `length` task that eliminates one particularly difficult pattern from the test data that occurred in 80.2 % of the examples there and not at all in the train set. Across all models, we see the accuracy rise dramatically upon making this fix.

The Ptr-Bahdanau architecture overall appears to behave quite similarly to its non-pointer counterpart, whereas the Ptr-Luong architecture exhibits vastly different behavior from the other three attention-based architectures. To further explore this difference, we show in Figure 1 the perfor-

Table 1: Final test accuracy in percent after training for 10,000 iterations. Shown are the mean and standard error of the mean (SEM) across 10 trials with different random initializations. The LB results are from Lake & Baroni (2018). LB-OB stands for the "overall-best" architecture: a 2-layer encoder-decoder LSTM with 200 hidden units and no attention. LB-Best is the best result reported across all the architectures they tested on the given task. *For the simplified_length task introduced in this paper, we show the accuracy achieved by our own reimplementation of LB-OB.

| Task | LB-Best | LB-OB | Bahdanau | Luong | Ptr-Bahdanau | Ptr-Luong |
|---|---|---|---|---|---|---|
| addprim_turn_left | **90.3** | **90.0** | **91.7 ± 2.9** | **89.9 ± 6.5** | **91.7 ± 3.6** | 66.0 ± 5.6 |
| addprim_jump | 1.2 | 0.08 | 3.7 ± 0.5 | 6.6 ± 1.4 | 3.3 ± 0.7 | **14.0 ± 2.8** |
| length | **20.8** | 13.8 | 15.7 ± 1.1 | 15.3 ± 0.7 | 13.4 ± 0.8 | 16.8 ± 0.9 |
| simplified_length | - | 83.9* | 76.7 ± 1.2 | 79.1 ± 2.7 | 69.4 ± 5.4 | **88.7 ± 3.8** |
| simple_p1 | - | ~5 | **81.3 ± 1.8** | 79.3 ± 1.0 | **84.1 ± 2.2** | 57.5 ± 1.9 |
| simple_p2 | - | ~54 | **98.6 ± 0.3** | 95.0 ± 0.6 | 96.0 ± 1.6 | 92.0 ± 0.9 |

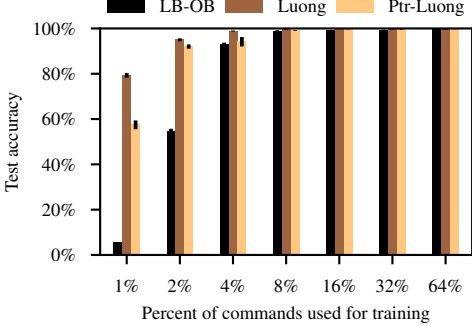 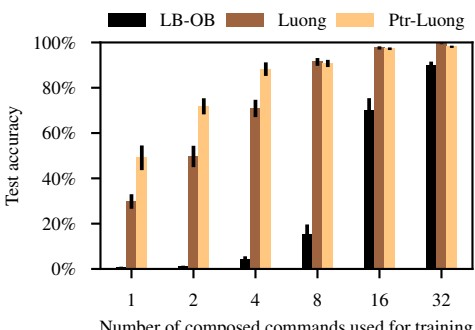

Figure 1: Test accuracy on *left:* simple_pX and *right:* addprim_complex_jump_numN tasks. The LB-OB results are taken from Figures 3 and 5, respectively, in Lake & Baroni (2018).

mance of the baseline, Luong, and Ptr-Luong models on two sets of tasks that measure data efficiency. In the case of identical train and test distributions (left), the Luong architecture is more data-efficient, especially at the very low end of using only 1 % of data for training. However, in the case of learning a new verb jump from only a few examples (right), the Ptr-Luong architecture performs much better when there are only very few samples.

The ambiguous benefit of Ptr-Luong versus Luong as well as the qualitatively different behavior between the Luong and Bahdanau attention types were surprising to us and we hope to explore these phenomena further in future work.

## 3 VARIANCE, ENSEMBLING, AND THE POTENTIAL OF IMPROVED SELECTION

To explore the variance due to the random initialization as well as the robustness of test accuracy to the amount of training, we trained 100 Ptr-Luong models with different random seeds on the addprim_jump task, and examined the test set predictions after every 1,000 iterations. The results are shown in Figure 2. The test accuracy shows extremely high variance across random seeds, peaks after around 2k iterations, and then rapidly declines.

In some sense, the presence of so many high-performing outliers is promising. If we had a good selection criterion, a viable strategy for achieving high performance on such tasks would be to simply train a lot of identical models and select the best one. In this case, the best-performing model at the best time achieved a remarkable 82.3 % accuracy.

Absent a selection criterion, we can still attempt to take advantage of this variance through ensembling. We try two variants, in both of which we select the answer that received the most votes. In the first variant, we fix a training amount (such as 3k iterations), and ensemble all the models at that

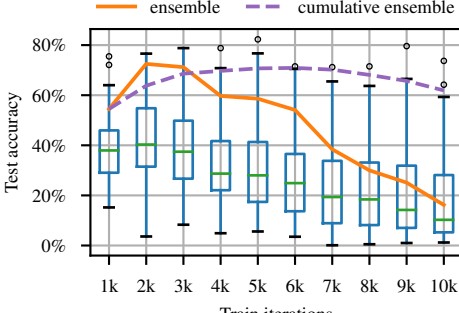

| Task | Mean | Ens. | Best |
|---|---|---|---|
| addprim_turn_left | 55.5 | 87.1 | 100.0 |
| addprim_jump | 16.9 | 16.2 | 82.3 |
| length | 17.3 | 19.8 | 40.1 |
| simplified_length | 84.6 | 86.7 | 100.0 |
| simple_p1 | 55.4 | 75.3 | 78.1 |
| simple_p2 | 90.7 | 95.6 | 97.3 |

Figure 2: *Left:* Box plot showing the distribution of test accuracy in percent by amount of training for 100 Ptr-Luong models on the `addprim_jump` task, as well as the accuracy for two ways of ensembling these models. *Right:* Mean, ensemble, and best test accuracy for 100 Ptr-Luong models.

stage of training. In the second variant, cumulative ensembling, we again fix a training duration, but ensemble all models up to and including at that training amount (e.g. the 100 models after they were trained for 1k, 2k, and 3k iterations). The results are shown in the left of Figure 2. Especially the cumulative ensembling is seen to greatly stabilize the test accuracy and is outperformed only by a handful of outliers. However, this is very particular to this task. As shown in the right of Figure 2, non-cumulative ensembling the fully trained models leads to above average performance on all the other tasks, with gains of around 20 % on `addprim_turn_left` and `simple_p1`. Still, there is much room for improvement: On the tasks with train-test distribution mismatch, ensemble performance is far below that of oracularly selected best single models with early stopping. Indeed, two of the tasks were solved perfectly by such models.

## 4 DISCUSSION

Unsurprisingly, our findings indicate that architectural inductive bias can indeed have a large impact. It could be interesting to take this further and train potentially even more compositional architectures such as Neural Programmer Interpreters (Reed & De Freitas, 2016; Cai et al., 2017). However, without extra supervision like execution traces, it seems unlikely that even these models would learn rules perfectly the way a human programmer might.

Instead, we would argue that the architectures studied here are in many cases already more than sufficiently compositional. When training many models with different random initializations, we observe that a number of them already achieve very good results, sometimes even solving the tasks perfectly. The problem is that (a) this does not happen consistently, and (b) even when it does, we do not have a non-oracular means of knowing it. In that sense, rather than pursuing more systematic architectures, it may be more fruitful to research how to make standard architectures more systematically achieve their full potential through selection or regularization:

- **Selection:** Traditional early-stopping and hyperparameter tuning methods based on train/dev/test splits do not apply to many of the tasks due to the train and test distribution mismatch. What can we substitute?

- **Regularization:** Can we regularize the train loss or the gradient descent procedure itself to more consistently arrive at the more compositional models that generalize better? Can we stabilize the test accuracy and avoid overfitting?

Overall, although the SCAN tasks are synthetic and small, we think they constitute a very useful initial test bed for new research ideas that aim to improve selection in train and test distribution mismatch scenarios or to improve generalization through encouraging pattern learning over memorization.

ACKNOWLEDGMENTS

We thank Yuanpeng Li and Jianyu Wang for helpful discussions.

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

## A    THE SCAN TASKS

The SCAN tasks are synthetic translation tasks, where input commands from a finite language generated by a simple grammar are to be translated into output action sequences. This is best illustrated by some examples:

| jump thrice | $\implies$ | JUMP JUMP JUMP |
| turn left after walk twice | $\implies$ | WALK WALK LTURN |
| run around right and turn left thrice | $\implies$ | RTURN RUN RTURN RUN RTURN RUN RTURN RUN LTURN LTURN LTURN |

For a full description of the SCAN language, we refer to Figures 6 and 7 in the appendix of Lake & Baroni (2018). For our purposes, it is most important to note that the verbs `jump, look, run, walk` behave in the same way, as do the directions `left, right`.

This small language is useful for testing whether models learn patterns or simply memorize data. Thus, if a model has learned the translation rule `X thrice` $\implies$ `[X] [X] [X]` and it has

learned `jump` $\implies$ JUMP, it should be able to compose this knowledge to correctly predict `jump thrice` $\implies$ JUMP JUMP JUMP.

We provide here a brief description of the tasks. Except for one task described below, we used the exact datasets made available at `https://github.com/brendenlake/SCAN`. We also follow the naming conventions for the tasks used in the website.

- `addprim_turn_left`: Among commands involving `turn left`, only the *primitive* example `turn left => LTURN` is included in the training data. (Note that `left` may still occur in combination with verbs other than `turn`. For example, `walk left thrice` is still in the training data.) The test set consists of all the *composite* examples involving `turn left`, such as `walk around right and turn left twice`.

- `addprim_jump`: Among commands involving `jump`, only the *primitive* example `jump => JUMP` is included in the training data. The test set consists of the remaining commands involving `jump`.

- `addprim_jump_complex_numN`: Like `addprim_jump`, but $N$ randomly selected composite commands involving `jump` are additionally included in the train set.

- `length`: Commands are split between train and test according to whether the output sequence has $\leq$ or $>$ 22 tokens, respectively.

- `simplified_length`: The train set is the same as for `length`, but we use a subset of the test data described below.

- `simple_pX`: Only `X` percent of the full dataset is used for training, the rest for testing.

The task `simplified_length` is not part of the original paper. We created it ourselves as we found that the pattern `[walk | jump | run | look] around [left | right] thrice` was completely missing from the training data of the `length` task but occurred in 80.2 % of the test commands. To test whether, aside from this particular pattern, models learn to properly generalize from short to long output sequences, we kept the same train set, but removed those 80.2 % of samples from the test set.

## B  MODEL DETAILS

We use the code base of Luong et al. (2017), which implements several state-of-the-art sequence-to-sequence models for neural machine translation (NMT) in TensorFlow. The table below lists the hyperparameter settings we used. (For hyperparameters not listed below, we used the default setting.)

These choices were made based on some preliminary experiments. Some explanation:

Table 2: Hyperparameters used for the experiments.

| Parameter | Values |
| --- | --- |
| num_layers | 2 |
| num_units | 768 |
| unit_type | lstm |
| encoder_type | bi |
| attention_architecture | standard |
| attention | scaled_luong \| bahdanau |
| pass_hidden_state | false |
| optimizer | adam |
| learning_rate | 1e-3 |
| dropout | 0.5 |
| num_train_steps | 10000 |
| batch_size | 128 |
| tgt_max_len_infer | 50 |

- The chosen settings for `pass_hidden_state`, `encoder_type`, and `attention_architecture` were found to matter a lot. Our selected settings were far superior to other options.

- Among attention types, `luong` was found to behave similarly to `scaled_luong`, and `normed_bahdanau` was found to behave similarly to `bahdanau`. So we picked only one representative of each type for our experiments.

- We use the same optimization and dropout settings as Lake & Baroni (2018), but we do not use teacher forcing and instead of 100,000 train steps of batch size 1, we use 10,000 train steps of batch size 128. The number 10,000 was a loose upper bound we observed for various architectures to converge on all tasks in preliminary experiments.

- Due to computational constraints and the difficulty of creating good development sets for these tasks, we did not tune the numerical parameters like `num_layers` and `num_units`. We just picked reasonable values based on a few preliminary experiments. Some further improvements could likely be obtained by tuning these, but we do not expect this to change our qualitative conclusions.

- Setting `tgt_max_len_infer` to 50 is just a technical fix. Without this, the code tries to infer a maximum target sequence length based on the length of the source sentence. For the SCAN tasks, this heuristic comes up with values that are too small.

## C  ATTENTION TYPES

Following Britz et al. (2017), we consider two attention mechanisms. In the Bahdanau variant (Bahdanau et al., 2015), the unnormalized attention score between an attention key $h_j$ (an encoder output) and an attention query $s_i$ (a decoder state) is calculated as:

$$\hat{a}_{ij} = \langle v, \tanh(W_1 h_j + W_2 s_i)\rangle. \tag{1}$$

where $v, W_1, W_2$ are trainable parameters. In the Luong variant (Luong et al., 2015), we instead use

$$\hat{a}_{ij} = \langle W_1 h_j, W_2 s_i\rangle. \tag{2}$$

The context vector $c_i$ is obtained by first normalizing the attention scores across $j$, and then extracting a weighted average from the values sequence:

$$c_i = \sum_j a_{ij} V_j, \tag{3}$$

$$a_{ij} = \underset{j}{\mathrm{softmax}}(\hat{a}_{ij}). \tag{4}$$

In the case of regular attention, the values vectors are again the encoder output states, $V_j = h_j$. In the pointer variant, the values vectors are the input embedding vectors.

## D  POINTER NETWORKS

Pointer networks were first introduced in Vinyals et al. (2015) to solve sequence-to-sequence problems where the output vocabulary is unknown, but each output token occurs in the input sequence. The idea is to use attention to select which of the input tokens to output at each time step. Originally applied to more combinatorial problems like the Traveling Salesman Problem, variations of this technique have since also proved useful on tasks like the Stanford Question Answering (SQuAD) dataset (Rajpurkar et al., 2016; Wang & Jiang, 2016).

In our case, the output tokens do not correspond directly to the input tokens, but there is a fairly simple mapping between them. To still be able to take advantage of the inductive bias of pointer networks, we modify the usual attention mechanism to select values from the input embedding. The decoder then uses these position-independent input embeddings to predict an output token.

The motivation for using pointers is as follows: If the model has learned the mapping `X` $\implies$ `[X]`, then a pointer-based decoding of a pattern like `X thrice` would need only to point to the first input position three times during decoding, independent of which particular token `X` is. Thus, using

pointers should help disentangle learning of the input to output token mapping from learning how to decoding particular sentence patterns. Intuitively, we would expect this inductive bias to improve data efficiency and generalization.

We implemented this in the code base of Luong et al. (2017) by modifying `attention_mechanism._values` to point to the input embedding rather than the encoder outputs.

## E  EXAMPLES OF ENSEMBLE PREDICTIONS

We show here some examples of predictions by the cumulative ensemble of 1000 Ptr-Luong models on the `addprim_jump` task.

The following are some examples of inputs on which the ensemble makes *correct* predictions, sorted by number of votes.

| votes | input |
|---|---|
| 917 | jump and turn opposite right twice |
| 916 | jump and turn left twice |
| 915 | jump and turn opposite left twice |
| ⋮ | ⋮ |
| 908 | turn opposite right twice after jump |
| ⋮ | ⋮ |
| 680 | jump opposite left after jump |
| ⋮ | ⋮ |
| 36 | jump opposite left thrice after jump around right thrice |

In contrast, here are some examples of *incorrect* predictions, sorted by number of votes.

| votes | input | prediction |
|---|---|---|
| 1000 | jump and jump twice | JUMP |
| 1000 | jump thrice after jump | JUMP |
| 1000 | jump and jump thrice | JUMP |
| 999 | jump after jump | JUMP |
| 999 | jump twice after jump | JUMP |
| ⋮ | ⋮ | |
| 935 | walk and jump thrice | WALK |
| 932 | run and jump thrice | RUN |
| 932 | jump after walk | WALK |
| ⋮ | ⋮ | |
| 31 | jump opposite right thrice after jump around right thrice | RTURN repeated 18 times |

One intuition for why ensembling could be so effective is that, when many models agree, they do so because they have learned the same correct underlying rule, whereas when they disagree, it is for more random reasons. As is evident from the examples above, that is not the case here. The ensemble is also often systematically incorrect. For example, all 1,000 models agreed that the command `jump and jump twice` should be mapped to simply `JUMP`. This pretty much rules out a semi-supervised approach, at least in the naive form where we try to use the ensemble's most confident test set predictions to augment the train set.

