# OpenReview forum: "More systematic than claimed: Insights on the SCAN tasks"
_ICLR.cc/2018/Workshop — Reject_

### Official Review · AnonReviewer3 · 2018-02-23
**As unsystematic as claimed**

**Rating:** 4
**Confidence:** 4

**Review:**

This extended article presents experimental work on the recently introduced SCAN tasks. The article ostensibly reports two main findings: 1) standard seq2seq architectures can do much better on these tasks than the basic models used in the original paper; 2) there is much variance in performance, such that the best model across random initializations even achieves 100% accuracy on a task where mean performance is at 17% (!).

Point 2) is very interesting. I think it would be worth, by itself, a more extended study. The authors might also want to take a look at https://arxiv.org/abs/1802.06467, which makes a similar point.

However, I failed to find support for 1) in the reported experiments. Unlike I am misinterpreting the results reported in Table 1, the only considerable improvement that the authors report pertains to the addprim_jump task, where their best model attains 14% accuracy. Now, this is indeed better than the originally reported 1% accuracy, but it is absolutely not justifying the "more systematic than claimed" pun in the title! If anything, the results in this table justify the opposite conclusion: even after testing various state-of-the-art models, SCAN is still far from being "cracked". The systematicity challenge still stands.

I am not sure about what the simplified_length experiment should demonstrate. To me, it shows that, by removing the most challenging cases from the SCAN length test set, performance greatly improves. Is this surprising? What should we learn from it?

Note that the original Lake & Baroni paper was rejected, so the citation should be of the unpublished arXiv manuscript.

To summarize, I found the second part of the article interesting, and it would be great to see it extended. However, I am giving a low score because the title and the claims made in the first part seem actively misleading.

---

### Official Review · AnonReviewer2 · 2018-03-09
**Some intriguing results but more a pile of puzzling experiments than a result**

**Rating:** 4
**Confidence:** 3

**Review:**

Summary:

This paper revisits the SCAN tasks  introduced in a (rejected, invited to workshop track) ICLR 2018 paper, which convert simplified English to a simple formal semantics action sequence and suggests that sequence models can do them rather better than the results reported in the original paper (Lake and Baroni 2018). They emphasize model selection, regularization and ensembling.

Novelty:

None really, but some interesting first cut results on this new dataset

Clarity:

Clear enough on what was done, not necessarily on what it means.

Significance:

Some interesting puzzle, not a lasting result

Quality:

Written up well enough but this seems a progress report on early results. Maybe that's what a workshop paper is meant to be, but this feels maybe too early.


Pros:

- If the Lake and Baroni paper is being presented at the ICLR 2018 workshop track, this would be a nice complement: It not only could generate debate, but shows that you can do considerably better than Lake and Baroni report in their paper.

Cons:

- This paper feels like someone who has done their first bunch of experiments and has some sort of interesting results in a we-don't-really-understand-what's-going-on way, but there isn't any new model, or new understanding or clear answer to the major puzzles. I suspect that the work would benefit from further development.

- Although the numbers change, the basic character of L&B's results/claims don't change AFAICS: An LSTM (+attention + ptr) works for a "mix and match" case like addprim_turn_left but not for a "systematic composition required" case like addprim_jump.

---

### Official Review · AnonReviewer1 · 2018-03-09
**Interesting new results on the SCAN task, not sure what to conclude**

**Rating:** 6
**Confidence:** 4

**Review:**

This paper presents the results of two experiments conducted on the SCAN data in response to Lake & Baroni’s yet unpublished, “Still not systematic…” paper (which in its latest version is now titled, “Generalization without systematicity…”) The first experiment examines the impact of two types of attention (additive or Bahdanau attention versus multiplicative or Luong attention), with and without a modification to have attention behave more like a pointer network. The second experiment looks at the impact of ensembling. As is apparently normal for the ICLR workshop track, this is an extremely dense 3-page paper that relies very heavily on its large appendices.

Pros:

Some of the results are interesting, and imply that Lake and Barroni’s hyper-parameter search was suboptimal.

Cons:

The most interesting results of Lake and Barroni still stand, more or less unchanged by this paper. It is difficult to tell what we should take away from this paper.

Clarity:

Overall, this paper is paper is fairly clear, though I could live without the structure that this format seems to necessitate, where the summary of results comes first, and all details of the model and experiments come second in the appendix. This includes important experimental details, like how to interpret cryptic experiment names like “simple_p1”.

For replication, in Appendix B, instead of saying, “we do not use teacher forcing”, it would be better to say what you do use.

I think it would also be helpful to explain the reasoning behind ensembling across both time and across different initializations. Generally, people ensemble over time only to approximate ensembling across seeds. Furthermore, the figures seem to indicate a benefit of ensembling only earlier in time - which seems to strangely conflate the value of ensembling with the value of early stopping.

Quality & Significance:

Unlike the authors, I was mostly unimpressed by the jump from 0.08% accuracy to 14% accuracy in addprim_jump task using pointer networks. As the authors point out, even though it is a big relative improvement, we are still nowhere near to solving the problem. Furthermore, performance on other tasks is sacrificed for this gain. So I’m not sure I agree with the statement that, “architectural inductive bias can indeed have a large impact.”
Likewise, the ensembling experiments are not particularly surprising, though I did like the observation in Appendix E that for some source sequences the ensembles can be very consistently wrong.

What I did find interesting were the big jumps on all of the various low-data tasks (simple_p* and the “number of composed commands used in training” experiments), which appear to have happened for all of the tested attention types (pointer or no). I feel like some more space needs to be devoted to why the author’s Bahdanau attention system is able to perform at 80% accuracy when Lake and Barroni’s appears to be well under 5%. The authors do a good job of listing the differences in their training regime (mini-batches, teacher forcing, etc), but they don’t devote any space to describing which differences led to such a dramatic improvement.

The authors’ catch regarding Lake and Barroni’s length experiment is also worthwhile, but it is more something one would hope to have appear in errata for the original paper.

Overall grade

I’m not sure if this paper adds enough to the work of Lake and Barroni to get too excited. As I suggested above, by far the most significant result in my eyes is that the quality of paper’s Bahdanau attention results implies a flaw in the original work’s parameter search, which in turn impacted its ability to do well in low-data scenarios. But without an explanation for what training or architecture difference was crucial for this jump, I cannot get too excited about that either.

Overall, I wonder if the most important statement in the paper isn’t the “Selection” bullet in the Discussion. Because of their inherently one-shot nature, these experiments don’t lend themselves to development-based early-stopping and hyper-parameter tuning. I feel like until we have a good answer to the selection problem, we will continue to see strange results like these, where architecturally very similar system do dramatically better or worse than reported in the literature.

---

### Author Response · Authors · 2018-03-21
**Some clarifications**

First, there seems to be criticism that the paper is not "finished". We do not disagree. We viewed the paper more as a combination of position paper and preliminary insight report. This felt appropriate for a workshop submission, but it is of course a judgment call and we respect the decision. We do think it could have generated useful discussion and constructive debate for those interested in compositionality and in designing improved tasks for research in this area.

Second, there seems to be an unintended interpretation of this paper as a strong "us vs them" sort of attack on the original paper (Still not systematic after all these years). This can perhaps be blamed on the titles as suggested by reviewers. But any reader of our paper will see that we think the SCAN tasks raise interesting issues, and we are mostly highlighting topics that seem worthy of further pursuit. This should be seen as more constructively complementary, rather than adversarial.

To elaborate on our alternate viewpoint on the interestingness of the SCAN tasks:

Parsing synthetic languages with regular grammars is not a particularly interesting ends in itself. We already know how to do this, even without machine learning. If we use domain-specific architectures, we can even achieve perfect accuracy with machine learning: See, e.g., https://arxiv.org/abs/1706.01284 (Towards Synthesizing Complex Programs from Input-Output Examples). Similarly, it is probably not too hard to come up with compositional architectures that can solve the SCAN tasks perfectly - but that may tell us preciously little about real-world challenges for machine learning.

The more interesting issue raised, in our opinion, is that of memorization vs rule learning. As we show, some of the SCAN tasks seem to be excellent test beds for this. It is very easy to achieve 100% train accuracy on the tasks, but the test accuracy can show extremely high variance. The natural hypothesis is that systematicity is not merely a consequence of the architectural choice, but also very strongly of how the architecture is trained.

Developing regularization or optimization techniques that can favor rule learning over memorization seems like a very timely topic that could have major implications for other deep learning applications as well.

In support of this, let us mention that, since submitting the workshop paper, we have found that through different optimization algorithms we can routinely achieve ~90% accuracy with low variance on the addprim_jump task with the exact same architecture (Ptr-Luong) used in this workshop paper.

Finally, the paper one reviewer suggested - Memorize or generalize? Searching for a compositional RNN in a haystack - has some similar findings and we second the recommendation for people interested in this topic. Note that we were not made aware of this paper and it did not appear on arXiv until after we had already submitted our workshop paper.

---

### Decision · Program_Chairs · 2018-03-20
**ICLR 2018 Workshop Acceptance Decision**

**Decision:**

Reject

**Comment:**

Based on the reviews, this paper has not been accepted for presentation at the ICLR workshop. However, the conversation and updates can continue to appear here on OpenReview.